# RoboTwin Metaverse Platform for Robotic Random Bin Picking

Cheng-Han Tsai [1,2], Eduin E. Hernandez [2], Xiu-Wen You [2], Hsin-Yi Lin [2] and Jen-Yuan Chang [1,3,*]

1 Department of Power Mechanical Engineering, National Tsing Hua University, Hsinchu 30013, Taiwan; lancetsai@itri.org.tw

2 Mechanical and Mechatronics System Research Labs (MMSL), Industrial Technology Research Institute (ITRI), Hsinchu 310401, Taiwan

3 Mechanical & Computer-Aided Engineering, National Formosa University, Yulin 632, Taiwan

* Correspondence: jychang@pme.nthu.edu.tw; Tel.: +886-3-574-2498

**Abstract:** Although vision-guided robotic picking systems are commonly used in factory environments, achieving rapid changeover for diverse workpiece types can still be challenging because the manual redefinition of vision software and tedious collection and annotation of datasets consistently hinder the automation process. In this paper, we present a novel approach for rapid workpiece changeover in a vision-guided robotic picking system using the proposed RoboTwin and FOVision systems. The RoboTwin system offers a realistic metaverse scene that enables tuning robot movements and gripper reactions. Additionally, it automatically generates annotated virtual images for each workpiece's pickable point. These images serve as training datasets for an AI model and are deployed to the FOVision system, a platform that includes vision and edge computing capabilities for the robotic manipulator. The system achieves an instance segmentation mean average precision of 70% and a picking success rate of over 80% in real-world detection scenarios. The proposed approach can accelerate dataset generation by 80 times compared with manual annotation, which helps to reduce simulation-to-real gap errors and enables rapid line changeover within flexible manufacturing systems in factories.

**Keywords:** metaverse; auto annotation; robotic random bin picking

## 1. Introduction

In today's manufacturing industry, vision-guided robotic picking systems play a vital role in automating various tasks. However, achieving rapid changeover and adaptability for diverse workpiece types has remained a significant challenge. Traditional approaches require constant redefinition of vision software and manual annotation of datasets, leading to time-consuming and tedious processes. To address these limitations, this study introduces a novel approach that leverages the capabilities of the RoboTwin, a metaverse platform developed by the Industrial Technology Research Institute (ITRI) of Taiwan.

The RoboTwin serves as a realistic virtual environment, replicating actual factory layouts, robotic cells, and workpieces. It offers a range of key features that revolutionize vision-guided robotic picking systems. Firstly, it incorporates physics engines that accurately simulate the real-world behavior of workpieces, enabling precise interaction with other objects. Secondly, it provides a modeling and automated AI tool capable of mapping real-world textures onto 3D CAD models of workpieces. This tool not only enhances the realism of the virtual environment but also automates the generation of annotated datasets, capturing pickable and unpickable points for each workpiece. Thirdly, the RoboTwin includes robotic models equipped with motion control algorithms for robotic arms and gripper reactions, ensuring realistic and optimized movements. Lastly, virtual reality (VR) modules are included to allow operators to immerse themselves in the virtual factory, which further facilitates testing and fine-tuning of the robotic picking system. To provide a visual demonstration of the RoboTwin system in action, we prepared a video that showcases its

features and capabilities: [RoboTwin (https://www.youtube.com/watch?v=AnX3v12v8y4 (accessed on 25 August 2022))].

Once highly realistic datasets are generated using RoboTwin, and the AI model is trained, they can be seamlessly integrated to form the FOVision system. FOVision represents an all-in-one robot vision module that combines machine vision, intelligent computing, and active guide light sources within a compact and highly flexible design. This versatile module can be effortlessly integrated with diverse robotic arms and machining equipment from different brands, enabling the convenient implementation of various functions including visual positioning, material handling, and assembly. By eliminating the need for complex programming, FOVision streamlines the deployment and operation of vision-guided robotic systems. To see the FOVision system in action and gain a better understanding of its capabilities, we created a video demonstration: [FOVision (https://www.youtube.com/watch?v=BuUc_1zTrrg (accessed on 17 August 2020))].

This research aims to address the existing technology gap by proposing a comprehensive solution that enables rapid workpiece changeover in vision-guided robotic picking systems. By utilizing the highly realistic metaverse scene provided by RoboTwin, the proposed approach offers several advantages. It significantly reduces the need for manual annotation by automating dataset generation, resulting in faster training of AI models. Moreover, it minimizes the simulation-to-real gap errors, enhancing the reliability and accuracy of the robotic picking system. The proposed approach enables rapid line changeover, meeting the demands of flexible manufacturing systems in today's factories.

To provide a comprehensive understanding of the research, this paper is organized as follows: Section 2 provides a literature review of relevant prior work and highlights the motivation behind this research by identifying the existing technology gap that this study aims to address. In Section 3, methods, materials, and procedures used by the RoboTwin system from dataset generation to the deployment of AI on the robot are described, as well as the FOVision. Calibrated experimental results are presented in Section 4, followed by a discussion in Section 5. Concluding remarks are presented in Section 6 with thoughts and recommendations for future work.

## 2. Background

According to the literature, researchers have shown that utilizing virtual scenes can be a replacement or comparison source to their real-world counterparts. As addressed by Rivera-Calderón et al. in [1], a lack of resources and accessibility to real-world scenes can be an issue in training robots in the digital twin. For instance, in [1], the visualization of a virtual robot that is controlled based on data captured from cameras in the physical environment was introduced to allow for the integration of knowledge and remote practice without the need for a physical robot, thereby increasing equity and inclusivity in assessment. Bansal et al. in [2] argued that advanced manufacturers often require dynamic changes on the factory floor to enable manufacturing. To address these requirements, they employed the ant colony optimization algorithm to program an industrial robot for obstacle avoidance and pathfinding for object picking and placement. In their work, the optimization was completed in a digital-twin environment, and after human inspection, the movements are transferred to a real robot, with all trials completed at a minimal cost. Borangiu et al. in [3] posited that although digital-twin systems consider the manufacturing process, they often ignore the gaps and discrepancies between the simulated and real-world cases. To cope with the problems, they proposed a smart manufacturing control model that ensures cost optimization and reality awareness of resource usage and suggested that this solution could enhance a digital-twin system by connecting it to the cloud and a management execution system (MES).

To summarize, the authors of [1–3] have provided valuable strategies and contributions for using virtual systems to support the digital-twin platform. However, the reported methods are not sufficiently robust to account for factory environments, primarily due to the following reasons:

- The level of realism in the virtual scene plays a crucial role in its corresponding generated value. If the simulated virtual scene is not realistic enough, the performance gap can be significant. In contrast, achieving a higher level of realism is more important than designing the platform itself, as the latter is readily available in the market.
- The movement of equipment is also an important factor to consider. Identifying and resolving issues in the virtual simulated scene before deployment in the real environment can greatly reduce deployment, tuning, and debugging time, as well as protect the equipment.

Research on using vision-guided robots in engineering applications such as surgical operations [4], where surgeons are assisted with automated surgical subtasks, has reported several challenges due to backlash, hysteresis, and variable tensioning in cable-driven robots. To address these issues, the authors in [4] proposed to employ visual feedback, referred to as deep intermittent visual servo (IVS), to estimate the robot's next-step action instead of using information from the robot's encoders. The results showed that IVS could achieve the highest published success rates for automated surgical peg transfer and was significantly more reliable than previous techniques when instruments were changed.

Lee et al. in [5] stated that operating a robot in a robotic bin-picking system's environment, where workpieces are randomly stacked in the crate, could be a challenging task for the robot because the posture of each workpiece is unknown and therefore could not be preprogrammed. To solve this issue, 3D cameras were used in their research to obtain the corresponding point cloud datasets to match with the CAD model to identify the posture. However, this solution was only useful for educational settings, not for real-world implementations in factories. In industrial settings, the role of operators involves the manufacturing of workpieces rather than their design. Consequently, access to the workpiece's CAD model is often unavailable to these operators. To address this limitation, an artificial intelligence system was introduced by the authors in reference [5]. This system aims to facilitate 2D object segmentation of workpieces, thereby enabling precise pixel-wise alignment of the bounding box. which could be used to determine whether the workpiece was pickable or not. Based on the segmentation, the model from [5] was able to estimate the normal direction of the targeted workpiece's surfaces in guiding the robot to conduct picking operations. Since the system does not require CAD models for matching, and high-resolution 3D cameras are not required, it significantly reduces the cost of common robotic random bin-picking systems. Additionally, with the use of AI to change and train the dataset, the aforementioned system can be easily transferred to detect another workpiece, which facilitates the need for rapid line change in factories.

Qiao et al. [6] tackled the long-standing and challenging inverse projection problem in computer vision by proposing the depth-aware video panoptic segmentation method. This method combines monocular depth estimation and video panoptic segmentation as a step toward solving the inverse projection problem. The method demonstrated that instance segmentation, although challenging, is useful in image processing. In another study [7], Shen et al. reported that a low-resolution grid is not sufficient to capture details, while a high-resolution grid would significantly increase the training. Discrete cosine transform (DCT) was proposed to encode the high-resolution binary grid mask into a compact vector. The DCT was demonstrated to be able to perform instance segmentation with high resolution and to achieve an AP50 score of 55.4, a higher score than with traditional Mask R-CNN.

Few-shot instance segmentation methods are promising in the case when labeled training data for novel classes are scarce. Using stored embedding vectors rather than images can effectively solve memory overhead problems [8]. The results are promising and can narrow the gap between non-incremental and incremental few-shot instance segmentation. The work presented by Chen et al. in [9] defines a new scale-aware search space where both image- and box-level augmentations are designed to maintain the scale invariance, resulting in higher AP scores for different models. Understanding the scene is a complex yet important task for a vision-based self-driving system. Researchers in [10] proposed a

part-aware panoptic segmentation method that combines instance and semantic segmentation methods. In deep learning, collecting a vast quantity of data is time-consuming and labor-intensive. To alleviate this issue, Yuan and Veltkamp [11] suggested utilizing a 3D environment simulator to provide photorealistic simulations by using a view synthesis module to support a flexible configuration of multimodal sensors. Their results showed that vision-based algorithms developed in the simulation can be transferred to real physical platforms without domain adaptation. In machine learning, if the labeled datasets are inaccurate, the annotations generated by the ensemble may lead to performance degradation of the trained model. To address this, Simon et al. [12] suggested a trained model that predicts the quality of the annotation from the degree of consensus between ensemble models. The results obtained in this study indicate that the process necessitates only 30% of the original data, and the rest can be replaced with automatically annotated data.

The research from [4–12] offers valuable insights and contributions regarding the utilization of AI-based vision systems for object detection without the need for reprogramming. Instead, these studies emphasize the retraining of the AI model using different datasets to achieve robust detection capabilities. However, there are still concerns specific to factory environments, including the following problems:

- The generation of images without annotations is insufficient for training the AI model, as manual annotations are still required;
- Manual annotation introduces instability, which directly affects the repeatability and reproducibility of the results obtained from the trained AI model;
- The image generation process only considers the object's posture, lighting intensity, and light colors. It does not consider factors such as lighting position, camera shooting posture, and realistic textures, all of which are crucial in providing diverse and realistic image datasets for training the AI model.

## 3. The RoboTwin System and FOVision

### 3.1. RoboTwin System

In this section, the model RoboTwin system is described in detail. The system includes the following four subsystems:

- Physics engines—These engines encompass physics models that are utilized to calculate the real-world reaction of the workpiece when it is physically picked or interacts with other objects.
- Modeling and auto AI annotation tool—This tool facilitates the mapping of real-world textures onto the 3D CAD model of the workpiece and automates the generation of datasets with annotations indicating the pickable and unpickable points of each workpiece.
- Motion control—The motion control aspect encompasses robotic models equipped with motion control algorithms for the robotic arms and gripper reactions.
- VR modules—These modules provide an immersive experience for operators, allowing them to immerse themselves in a realistic virtual factory environment. This enables testing and fine-tuning of the robotic picking system.

Additionally, in what follows, the FOVision system will be described, of which the system deploys the trained AI model into the real-world robot for visual guidance in picking and placing operations. It allows bridging the gap between virtual and real environments by accurately detecting and identifying issues in the virtual scene, and then implementing solutions in the real world, resulting in a more efficient deployment, tuning, and debugging process, as well as better protection for equipment.

### (1) Physics engines:

The proposed RoboTwin system goes beyond utilizing existing physics engine functions. It incorporates additional physics algorithms to enhance the realism and robustness of the simulation. Within the simulator, various functions have been developed to calculate and retrieve important parameters, such as the workpiece's material, surface roughness,

weight, motion speed, the kinematics of the vacuum gripper, and other crucial factors that significantly impact the success of the robotic picking system.

Similar to the work of Tuleja and L. Šidlovská [13,14], who demonstrated positive results in workpiece picking using multiple grippers, we developed and fine-tuned our algorithm based on physics models to optimize the performance and minimize the disparity between the simulated and real values. This is achieved through a comparison of predicted outcomes obtained from the simulation with actual measurements. By doing so, we ensure that the RoboTwin system offers accurate and realistic simulations of the factory environment and the robotic picking system, enabling efficient testing and fine-tuning before deployment in real-world applications. As an example, the equation for successful pressurized picking is expressed as follows:

$$p = \frac{4mt(\alpha \pm g)}{10^{-3}\pi\mu(10^3 d)^2},$$

(1)

where $p$ is the minimum air pressure required by the robot vacuum gripper to hold the workpiece; $m$ represents the mass of the workpiece; $\alpha + g$ is the net acceleration at the gripping point from both the acceleration and gravity, respectively; $\mu$ stands for the coefficient of friction; $t$ is a safety factor defined by the orientation of the gripper in grasping a workpiece; and $d$ is the diameter of the gripper. However, there are often gap errors that come from unaccounted variables and small imperfections in measurements. To address this issue, after running experiments using Equation (1) on both the simulation and real-world environments, this gap error can be reduced by fine-tuning the parameters and applying linear minimum mean square error (LMMSE) on the acquired data. Suppose the real measurement is expressed as $Y$ and the simulated one as $\widetilde{Y}$, one can improve upon the estimated value $\hat{Y}$ as a function of $\widetilde{Y}$ as follows:

$$\hat{Y} = g\left(\widetilde{Y}\right) = a\widetilde{Y} + b$$

(2)

where $a$ and $b$ are scalars to be determined. More specifically, the goal is to choose $a$ and $b$ such that the mean square error (MSE) of the above estimator is minimized through the following equation:

$$\text{MSE} = E\left[\left(Y - \hat{Y}\right)^2\right]$$

(3)

Since $\hat{Y} = \hat{p}$, Equation (1) now becomes

$$\hat{p} = a\frac{4mt(\alpha \pm g)}{10^{-3}\pi\mu(10^3 d)^2} + b$$

(4)

**(2)    Modeling and Auto AI Annotation Tool:**

It is crucial to accurately replicate the real-world environment within the simulator, particularly when it comes to the workpiece in the metaverse robotic random bin-picking system. The fidelity of the workpiece's geometry and texture directly impacts the realism of the virtual scene and can influence the gap error. For instance, if a simulator generates images without considering real-world characteristics, training an AI model with those images may result in overfitting unrealistic features and lead to failure when applied to real images. To address this challenge, we employed an RGBD camera to capture the workpiece's real-world texture and simultaneously obtained the 3D geometry point cloud for generating the workpiece's mesh. This allowed us to achieve proper UV mapping of the texture [15]. UV mapping enables the determination of the relationship between the 2D image pixel $(u, v)$ and 3D coordinates $(x, y, z)$ of the mesh using the following geometric equations:

$$u = 1/2 + \tan^{-1}(2(x,z))/2\pi \text{ and}$$

(5)

$$v = 1/2 + \sin^{-1}(y) / \pi, \tag{6}$$

where $u$ represents the horizontal coordinate, and $v$ represents the vertical coordinate. The resulting values of $u$ and $v$ fall within the range of [0, 1] and are used to determine the texture mapping for the corresponding 3D point. By combining real-world scanned point clouds with 3D-model-generated point clouds, The accuracy of the simulated workpiece as a representation of its real-world counterpart can be assured. This approach enables us to train the AI model using realistic data, thereby reducing the gap error and improving the performance of the robotic picking system.

The auto AI annotation tool proposed in this research performs the following three main tasks: (1) scene generation, (2) image capturing, and (3) auto annotations. For scene generation, the tool utilizes a domain randomization algorithm to control several parameters to randomize the reaction of the camera, the lighting, the crates, and the workpieces. This process introduces variations in the scene setup, enabling a more comprehensive training dataset. Once the scene is generated, the tool captures high-quality images of the scene with the workpieces positioned within it. These images, along with the details extracted from the simulator, serve as the input for subsequent annotation processes. The tool automates the annotation process by accurately labeling the workpieces within the captured images based on the specified labeling conditions. It performs instance segmentation and identifies and categorizes objects of interest within the image using details that can only be accessed from the simulator.

For this tool, a simulator with a physics engine and domain randomization was implemented, the latter of which can be defined using the following function:

$$D(P, Q, CP, LP, LI, LC, CR), \tag{7}$$

where each of the parameters and functions is defined as follows:

- Domain randomization ($D$): A function that takes multiple randomization parameters and initiates the scene randomly based on these values.
- Object posture ($P$): Random postures ($x$, $y$, $z$, $r_x$, $r_y$, $r_z$) for the workpieces so each of them has a distinct posture and surface facing the camera while being randomly stacked inside a crate. Here, $r_x$, $r_y$, and $r_z$ represent the orientation or rotation angles with respect to the $x$-, $y$-, and $z$-axis, respectively.
- Quantity ($Q$): Random quantities of workpieces for each simulation scene. Depending on the quantities of workpieces, scenarios such as stacking and occlusion can have a higher probability of occurrence.
- Camera posture ($CP$): Random position for the camera to enhance or reduce workpieces in the image.
- Lighting posture ($LP$): Random posture for the lighting for each simulation scene so the light reflections are different.
- Light intensity ($LI$): Random light intensities are employed to simulate the light decay intensity.
- Light color ($LC$): Random light red, green, and blue colors (R, G, and B) influence the lighting in the scene.
- Color rendering ($CR$): Different color renderings of the objects are generated from the changes in the light sources. Only the high general color rendering index (Ra) allows the objects to show their natural color.

It is important to clarify that the parameters provided serve as bounds and conditions for the randomization process, i.e., when setting $Q = [1, 5]$, it specifies a range of workpieces from 1 to 5 that can be included in each generated scene. These parameters define the variability and constraints applied during scene generation, allowing for flexibility and control over the characteristics of the generated scenes. By doing so, the tool can generate a diverse set of images with different lighting conditions, camera angles, and workpiece poses. This ensures that the AI model is trained on a diverse set of images and can perform

well in real-world scenarios with varying conditions. Furthermore, the use of automated annotation tools can save time and effort compared with manual annotation, allowing for faster and more efficient dataset generation for training AI models.

Like in the real-world scenario where only the easily accessible workpieces can be picked, we designed the annotations so the picking class for each workpiece would consider occlusion level and the visible pixel surface area seen with the camera. For example, workpieces on the top of the crate normally have a higher priority for being picked since they have less occlusion and higher pixel surface area visible to the camera in comparison to other workpieces. The inclusion of this annotation makes the picking easier for the robot. Normally, workpieces closer to the middle of the crate should have a higher priority than those closer to the walls as they can be considered obstacles, potentially increasing the difficulty in the picking operations. With the annotations, the proposed system can simulate real-world scenarios and allows the AI model to learn the difficulties of picking workpieces in different positions, both of which can improve the performance of the robot in real-world applications.

**(3)   Motion control:**

In a robotic picking system, vision detection and guidance serve as core functions, while path planning is essential for navigating environmental obstacles. Established algorithms like rapidly exploring random tree (RRT) [16] are commonly used for path planning in robotics, but our focus in this paper is on the areas that require further research and improvement.

In many factory settings, obtaining 3D CAD models of machines and workpieces can be challenging due to copyright concerns. This is particularly true when using older equipment, where manual teaching of the robot's desired posture becomes necessary. Without a CAD model to import into a simulator, automatic path generation is not possible, making manual teaching a crucial aspect of motion control in the factory environment.

To address this challenge, the proposed RoboTwin system enables the robot to operate effectively in factory environments, even when 3D CAD models of machines and workpieces are unavailable. This enhanced capability provides greater flexibility and adaptability in various factory scenarios.

**(4)   VR modules:**

To circumvent the time-consuming and tedious task of manually teaching the robot postures, as shown in Figure 1, we propose using a VR glass to link the simulator and the real world. This allows users to control the robot via a joystick in the virtual world, facilitating the teaching of the robot for postures both for the inspection and exploration of feasibility. Once confirmed, the posture point can be directly transmitted to the real-world robot, allowing for a more efficient and accurate teaching process. This approach not only reduces the time and effort required for manual teaching but also allows for greater flexibility and adaptability in the factory environment.

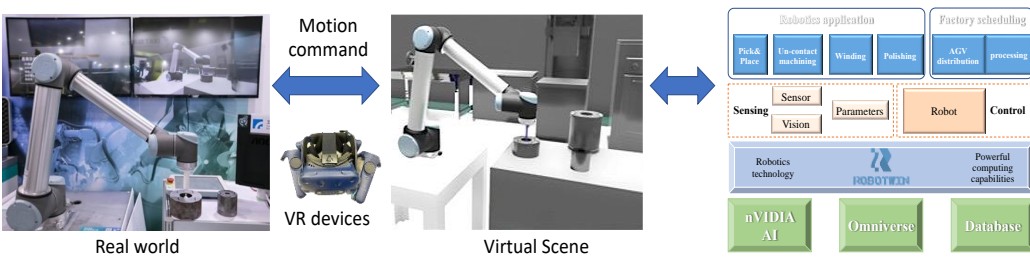

**Figure 1.** Platform structure of the VR modules with the RoboTwin system and the real-world robot.

*3.2. FOVision*

As illustrated in Figure 2, the FOVision system was adopted in the proposed system to verify the performance of the AI-trained model on synthesizing data by implementing

the path generated using the simulator and the VR modules. It deploys the AI model into a real-world edge computing system to detect the workpiece and calculate the path between the real robot gripper and the targeted pick-and-place locations [17]. The vision modules of the FOVision system include an edge computing unit, lighting, and an RGBD camera, all of which are integrated to connect the robot and the simulator via an ethernet hub. This allows for real-time visual guidance and detection of the workpieces, enabling the robot to efficiently pick and place the workpieces in the factory environment.

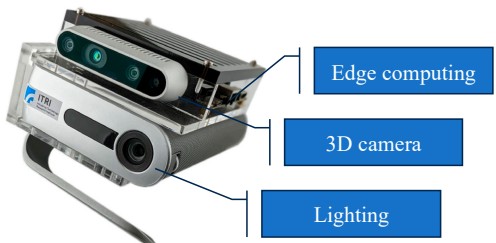

**Figure 2.** The FOVision system consists of an edge computing unit, a 3D camera, and a lighting unit developed by ITRI.

## 4. Results

To validate the performance of the proposed methods, we separate this section into two subsections, namely the RoboTwin system and the FOVision system, respectively.

### 4.1. RoboTwin System

4.1.1. Physics Engines

In the experiments, we used a robot equipped with a vacuum gripper to pick up and place standard weights of varying masses to assess the effectiveness of Equation (4). The experiments were conducted using the variables outlined in Section 3.1 and the values listed in Table 1 following the procedure described herein. We first picked up a weight of mass $m$, activated the vacuum gripper with the maximum allowable air pressure $p$ (i.e., 160 kpa as listed in Table 1), and then attached the weight to the gripper. We then gradually decreased the pressure until the weight dropped due to insufficient pressure, and recorded the corresponding $p$ value. This recorded value indicates the minimum pressure required to hold an object of that specific mass. We repeated this process with weights of increasing mass $m$ and compared the results from the real gripper to those from the simulator using the physics engines.

**Table 1.** Experiment parameters.

| Parameters | Value |
| :---: | :---: |
| $\alpha$ | 0 (Push the object toward the Suction Pad) |
| $m$ | 0.5–4 kg |
| $p$ | 10–160 kpa |
| $d$ | 0.04 m |
| $\mu$ | 1 (Suction pad must be parallel to the ground) |
| $t$ | {1, 4} (Safety factor introduced by authors in [14] to account for the orientation of gripper) |

The results from the aforementioned experiment are provided in Figure 3, where the solid blue line are values obtained from the gripper with measured pressure, while the

green and orange solid lines are the predicted values from the simulator using *t* values of 1 and 4, respectively. Note that *t* = 1 indicates no safety factor employed, whereas *t* = 4 indicates the safety factor is four times that of no safety factor applied. According to the experiment conducted on the real gripper and provided in Figure 3, when the mass is 1 kg, the gripper is not able to hold the standard weight if the air pressure provided falls below 25 kpa. However, from the simulations using a $t = 1$, this standard weight falls with 7.8 kpa or lower, and when $t = 4$, 31 kpa or lower as calculated from Equation (4). This indicates that the simulator's predictions are not accurate enough, and more realistic factors need to be considered. Such factors include the friction and deformation of the gripper and the workpiece. Additionally, the safety factor *t* may need to be adjusted to a higher value to account for these factors. Overall, these results indicate the importance of verifying the simulator's predictions with real-world experiments to ensure the accuracy and reliability of the picking system.

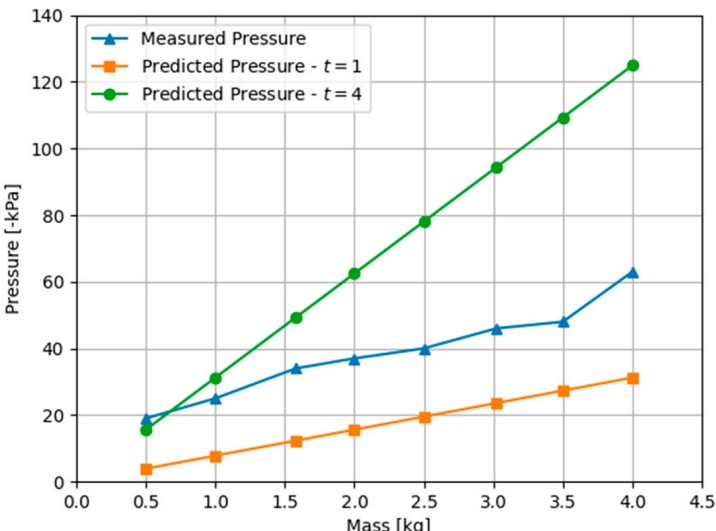

**Figure 3.** The relationship between gripper pressure required given a specified mass and constant parameters provided in Table 1 using the experimental results from the gripper with measured pressure and predicted pressure, either with a safety factor of *t* = 1 and *t* = 4 for the prediction.

As discussed in Section 3, there exists a gap between simulation and real-world results. To fulfill this gap, we optimized the results by adding parameters for tuning through the LMMSE algorithm. Optimizing the scalars in Equation (4) gives us $a = 1.619$ and $b = 10.400$ for $t = 1$ and $a = 0.405$ and $b = 10.400$ for $t = 4$, respectively, which reduces the mentioned error and provides comparable results. As shown in Figure 4, the dashed green and orange lines are the corrected prediction using the LMMSE. Note that this correction leads to the overlapping of both green and orange dashed lines, even with different *a* and *b* values. The loss provided in the description of Figure 4 was calculated using the MSE between the measured and the respective simulated pressure values. The predicted values with $t = 1$ yield a lower error rate than the predicted values with $t = 4$ in terms of their deviation from the real measured pressure, but both can achieve an even lower error rate through LMMSE. A detail worth noting is that the real measured pressure recorded was read directly from an analog pressure valve, so the recorded *p* value may not be precise. As such, a possible error region of ±7 kpa is provided by the shaded blue area in Figure 4. Furthermore, in these simulations, the smallest mass used was 0.5 kg due to challenges in determining the pressure measurements for lower mass values.

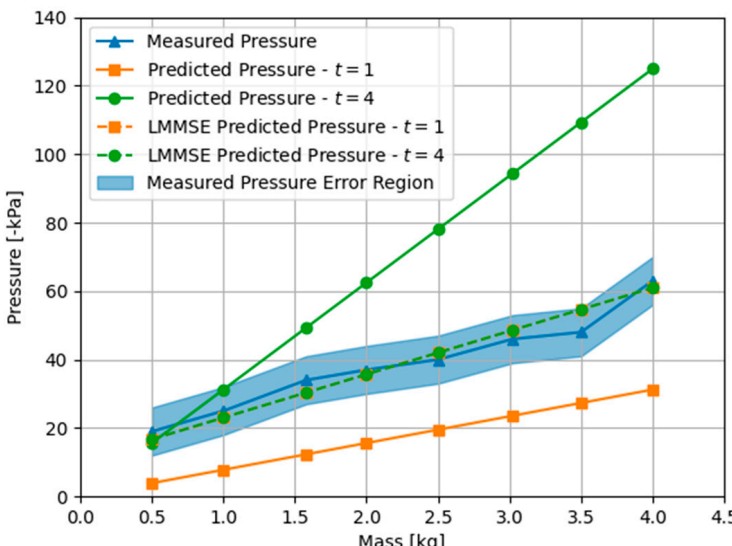

**Figure 4.** Gripper pressure required given a specified mass and constant parameters provided in Table 1. Using the experimental results for the gripper with measured pressure in blue solid lines, simulated pressure with $t = 1$ (MSE: 476.19) in orange, and $t = 4$ (MSE: 1538.68) in green for both solid lines. The dashed lines indicate LMMSE correction on the prediction (MSE: 10.39), and the blue shaded area indicates an error region on the recorded pressure due to imprecise readings on the pressure valve.

### 4.1.2. Modeling and Auto AI Annotation Tool

To capture the geometry and texture of the workpiece, we utilized a fixture equipped with a 3D camera. The camera is integrated into the fixture in a way that allows it to capture the workpiece's geometry and texture simultaneously, producing a data point cloud. This data point cloud is saved in the standard obj format, creating a functional model of the workpiece that preserves its textural information. Figure 5 provides an illustration of this process.

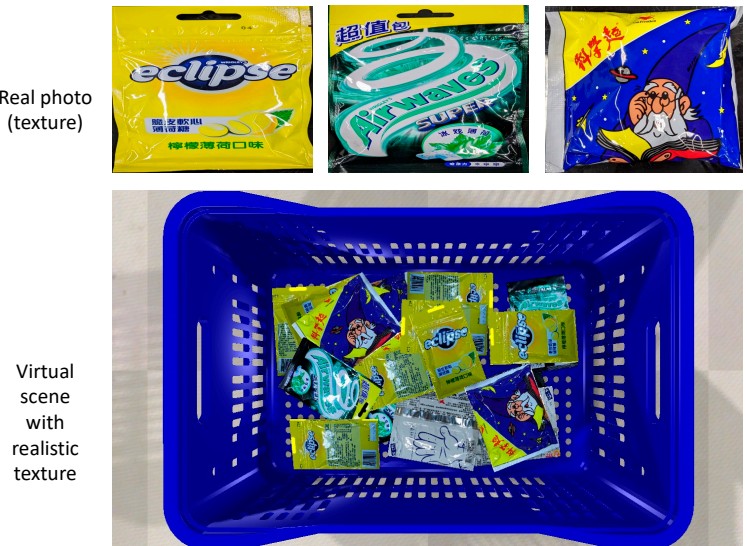

**Figure 5.** Modeling results showing real workpiece geometry and surface texture.

The proposed system incorporates a scanning apparatus to create a realistic model, which is then integrated into a simulator. This integration enables the utilization of domain randomization techniques to generate a wide range of diverse datasets for training the AI model. Figure 6 illustrates the simulated environment, showcasing different workpiece

quantities, postures, crate colors, lighting conditions, and postures. These variations are achieved by applying the same randomization parameters described in Section 3.1, resulting in diverse outcomes. This approach facilitates the collection of large and diverse datasets, which can be subsequently used for automatic annotation, as demonstrated in Figure 7, to further enhance the performance of the AI model.

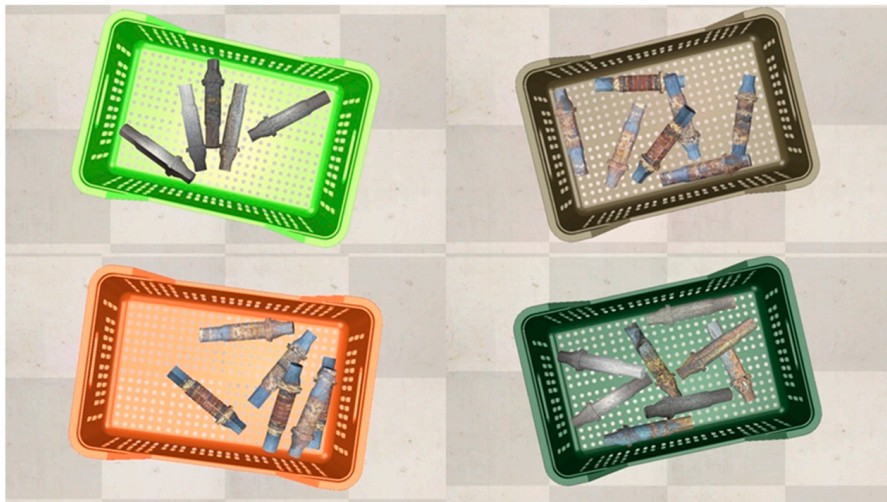

**Figure 6.** An example of domain randomization results using different parameters in the simulator.

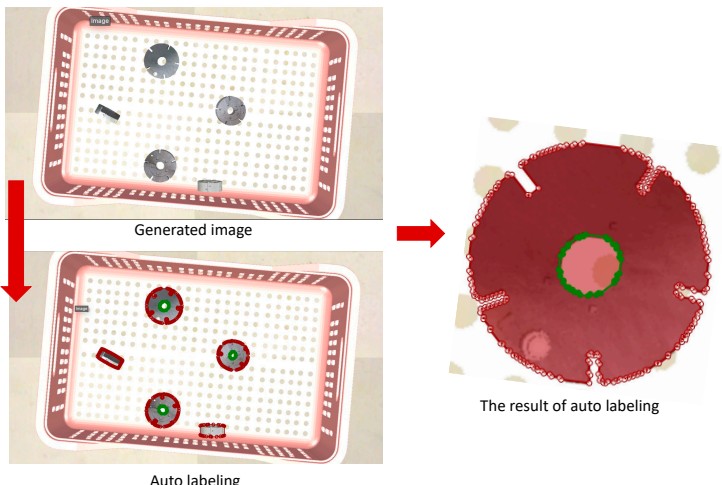

**Figure 7.** Auto labeling results in circular-like workpieces in the crate.

### 4.1.3. Motion Control and VR Modules

With the highly realistic textures and object responses in simulators equipped with physics engines, vision detection systems can be effectively utilized and tested within these simulated environments. This allows for the detection of workpieces and the generation of motion paths for robots, all directly from the simulation, as illustrated in Figure 8. To streamline this process, Figure 9 demonstrates the use of a VR device for efficient teaching and verification of these points. Through the VR device, the virtual world can be viewed via a virtual camera, and the controllers enable the operation of the virtual robot. Once the points and path are confirmed, the signal can be transmitted to the real-world robot, enabling it to perform the same movements, as demonstrated in the simulation, akin to a digital-twin system.

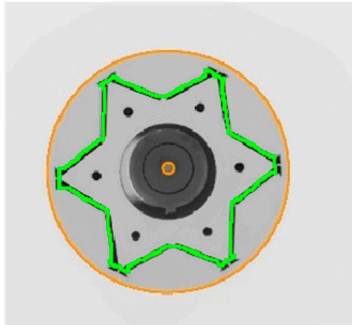
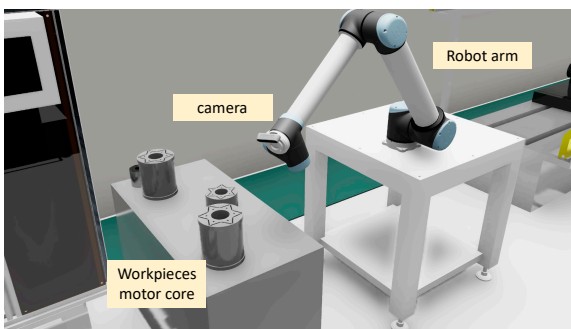

**Figure 8.** Results of the vision guiding robot in the simulated virtual world, where the green lines in the left image indicate the motion path for the robot generated using the proposed simulator to follow the star-like path.

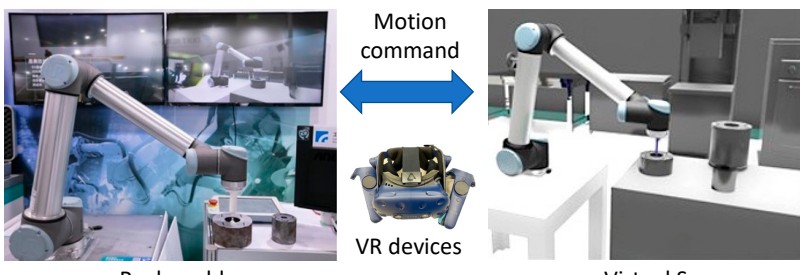

**Figure 9.** The connection between the real world and the simulated virtual world using the VR device and TCP/IP for streamlining data transmission.

*4.2. FOVision System*

In this experiment, we employed the synthetic dataset generated using the RoboTwin system to train the AI model and then deployed the trained model into the counterpart real system with the intent of detecting the real workpiece. Table 2 provides the detection results using YOLO3 with the bounding box mean average precision (mAP) for a threshold of 70%. Our results indicate that, with the synthetic dataset, a test accuracy of 71.3% can be reached, suggesting that the bounding box is precise enough for robot picking systems. Figure 10 shows the deployment results of this AI model on real-world workpieces. The comparative numbers tabulated in Table 3 demonstrate the necessity of automating the annotations. The manual part involved recruiting 20 students as participants to assess the time and accuracy, with the number of objects in each image being random to account for the inconsistency among personnel and provide a measure of validation accuracy. One can observe that the number of annotations generated using the proposed system outperforms human annotations by a factor of 80 for instance segmentation. As shown in Table 4, the successful picking rate for the RoboTwin system and the AI model is above 80%.

**Table 2.** Performance of vision detection from the simulator to real.

| Model | Train Size | Validation Size | Using Auto Tool | mAP70 |
|-------|-----------|-----------------|-----------------|-------|
| YOLO3 | 2700 | 300 | YES | 0.713 |
| YOLO3 | 2700 | 300 | NO | 0.423 |

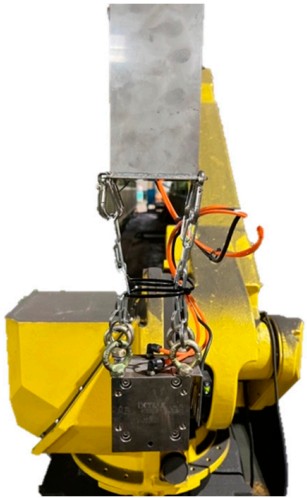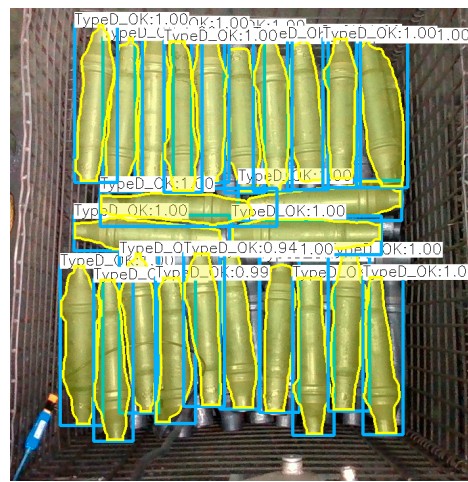

**Figure 10.** Photographs showing the deployment of the AI model trained on the synthetic dataset and tested in the real world. The image on the left is a robot with a gripper and camera in which we deployed the trained AI. The image on the right is the detection results using this AI model.

**Table 3.** Performance efficiency of auto annotation.

| Item | Annotation Type | Annotation Speed |
|---|---|---|
| Manpower | Bounding box | 60 img/hr |
| | Instance segmentation | 12 img/hr |
| RoboTwin System | Bounding box | 2000 img/hr |
| | Instance segmentation | 1000 img/hr |

**Table 4.** Picking the success rate of the robot using the RoboTwin system.

| Workpiece | Cycle | Success | Success Rate |
|---|---|---|---|
| Wrench | 500 | 400 | 80% |
| Drive shaft | 120 | 100 | 83% |

## 5. Discussion

As shown in Figure 4, after the inclusion of the additional parameters into Equation (1), the simulator can provide comparable outcomes to the real physical results measured. By finetuning the additional parameters, the difference between the real world and the simulation can be resolved. It is noted in Figures 5, 6 and 10 that using the real texture on the workpiece's 3D models can further enhance the realism of the annotated images used in the training and deployment of the AI model. This can in turn improve the detection of factual workpieces. The texture used in the simulation is crucial for the AI model because the features present in the workpieces, such as imperfections, cannot be generated using conventional computer-generated graphics. As indicated by Figure 6 and Table 1, it can be observed that the domain randomization method utilized for the dataset generation not only increases the number of images but also increases their diversity, which helps prevent overfitting in the AI during the training. The auto-labeling carried out using the RoboTwin system, as illustrated in Figure 7, demonstrates that the proposed system can be more stable and precise than human labeling. As the demonstrations shown in Figures 8 and 9 suggest, the proposed methods and system allow engineers and researchers to conduct experiments in a virtual world while still being able to deploy the solution in the real world. Such demonstrations showing the effectiveness of the proposed system and its contribution to enhancing existing knowledge are evidenced in Table 2 and Figure 10, showing a successful

transfer of the robot motion and vision detection from simulators with physics engines into the real world.

There are several factors that can influence the success rate of a robot picking system, including the accuracy of visual recognition, limitations of the robot's motion capabilities, and the design of the gripper. In this paper, we focused on verifying the effectiveness of visual recognition by comparing the success rate from virtual to actual scenarios using the same software functionality verification with the related robot and gripper available in our lab. However, the specific impact of the gripper and other mechanical components on the picking success rate will be a topic for future research and exploration.

## 6. Conclusions

This paper presents a solution for solving the challenges of using AI-based vision systems for robotic picking tasks in real-world factories. We proposed the use of a highly realistic metaverse scene for advanced tuning of the robot's movement and gripper's reaction, as well as the use of real textures on virtual 3D models for automatically generating a wide variety of annotated synthetic images to be used as a dataset for training the AI model. Through calibrated experiments with the proposed RoboTwin system, promising results were observed, showing an instance segmentation mean average precision of 70%, a picking success rate of over 80%, and the ability to generate datasets 80 times faster than manual annotation. The proposed methods and system can reduce the gap error between simulation and real-world environments, and provide a rapid line changeover to meet the demands of flexible manufacturing systems. The proposed RoboTwin system holds significant implications not only in engineering applications within smart manufacturing but also in advancing existing knowledge. It represents a notable step forward in the field of robotic picking, particularly in real-world factory environments.

**Author Contributions:** Conceptualization, C.-H.T.; methodology, C.-H.T.; software, C.-H.T., E.E.H. and X.-W.Y.; validation, C.-H.T., E.E.H., X.-W.Y. and H.-Y.L.; formal analysis, C.-H.T.; investigation, C.-H.T.; resources, C.-H.T.; data curation, E.E.H., X.-W.Y. and H.-Y.L.; writing—original draft preparation, C.-H.T.; writing—review and editing, E.E.H. and J.-Y.C.; visualization, C.-H.T.; supervision, J.-Y.C.; project administration, C.-H.T.; funding acquisition, C.-H.T. All authors have read and agreed to the published version of the manuscript.

**Funding:** This research was funded by Industrial Technology Research Institute (ITRI) with grant number M353C82210.

**Institutional Review Board Statement:** Not applicable.

**Informed Consent Statement:** Not applicable.

**Data Availability Statement:** Not applicable.

**Conflicts of Interest:** The authors declare no conflict of interest.

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
