# Peer review of "RoboTwin Metaverse Platform for Robotic Random Bin Picking"

_applsci, doi:10.3390/app13158779_

Round 1
Reviewer 1 Report
This paper proposed the RoboTwin metaverse platform for vision-guided robotic random-bin-picking, which can apply the simulation results of the virtual environment to the real world. The experimental results show that the above system has obvious advantages in instance segmentation, picking success rate and auto annotation. However, this paper does not explain every sub-function of the system clearly, and the innovation and contribution of this paper need to be further condensed. Some suggestions and concerns are as follows.
(1) The abstract part should further condense the work of this paper. In addition, since this article is about Robotwin, it is recommended to mention it in the abstract;
(2) The introduction part lacks the macro-background introduction of the article;
(3) It is suggested to elaborate the relationship between robotwin system and FOVision;
(4) In Section 3, the chapter Settings are not appropriate. It is suggested that the content of “Robotwin system” should be subsection 3.1 and the content of “FOVision” should be subsection 3.2;
(5) The "u" of formula 5 and the "v" of formula 6 are both equal to sinθcosφ, which is obviously incorrect;
(6) It is recommended to detail specific approaches of key technologies such as auto annotation and motion control;
(7) In Figure.2, it is recommended not to put the product manual directly in the article. Key parameters can be presented in tabular form;
(8) In Line 315, subsection3.1 is mentioned. But the variable mentioned in that line is not in subsection3.1 of this article;
(9) In Line 344, the experimental conclusion mentions "reliability", but the experimental results do not verify that the proposed method can improve the stability of the system. It is suggested that the author supplement relevant experiments;
(10) In subsection 4.1.1, please explain why the results of t=1 and t=4 should be corrected;
(11) Please check whether "a, and t=1" in Line 354 is correct;
(12) In Figure.5, the fourth picture doesn't look like a virtual photo;
(13) In Figure 6, it's better to include the specific parameters for each picture;
(14) Figure 7 is more like the result of image recognition. It is suggested that the author interpret the content of automatic annotation in Figure 7;
(15) YOLO2 is mentioned in subsection 4.2, but the corresponding results are not given in Table Ⅱ;
(16) Automated annotation is certainly faster than manual annotation, but I don't think manual annotation is as slow as in Table Ⅲ. It is recommended that the author explain how the manual annotation speed is obtained in Table Ⅲ. While speed is important, accuracy is even more critical. Therefore, it is suggested that the authors provide the accuracy rates of the two annotation methods for comparison to reach the conclusion “more stable and precise than human labeling” of Line 440-441;
(17) The conclusion “provide a rapid line changeover to meet the demands of flexible manufacturing systems” of Line 458-459 are not supported by corresponding experimental results. It is suggested that the author provide relevant experiments for verification.
Author Response
Thank you for the comments and suggestions. Kindly please see our responses in the attached file. Thank you!

Reviewer 2 Report
I would like to congratulate the authors for the very interesting paper.
Please consider some observations from my side:
- Eq.2 calculates the estimated value in function of the simulated value, not the real measured value is expressed in function of the simulated value, as stated in line 196.
- line 358 – based on what is taken the width of the “possible error region”?
- the caption of the Figure 4 (lines 362-367) is divided by page break
- line 410 – YOLO2 is mentioned, but in the result Table II is not present
- lt would be nice if in the 5th chapter would be some discussion about the picking success rate presented in Table IV. In my reading the picking is a success if the part is taken form the bin. It is unclear that the unsuccessful cases are due wrong segmentation, wrong picking point determination on the part, or lack of gripping/suction force.
- The two parts of the RoboTwin (Physics engines and Modeling and Auto AI Annotation tool) are presented somehow separately, having different use cases. To expose the system’s strength, I would present the segmentation and bin picking (determining the right p vacuum pressure) for the real workpieces from Figure 5. Or other real workpieces having more mass.
Author Response
Your comments and suggestions are much appreciated. Please kindly see our responses in the attached file.

Reviewer 3 Report
1. The title should be revised since the contribution of this manuscript is dataset generation for robotic bin-picking. Thus, the term 'platform' is not rigorous.
2. Some references for semantic or instance segementation should be cited, especially, some simulation-based dataset generation should be emphasized.
3. Theoretical contributions are not outstanding. Only an introduction of a system is presented. The author should summarize some main contributions.
The form. English should be improved throughout the manuscript. The form is often too colloquial. The style is often too verbose. The authors are invited to carefully check the manuscript in this sense.
Author Response
Comments and Suggestions from the reviewer and authors' responses.
Comment/Suggestion #1: The title should be revised since the contribution of this manuscript is dataset generation for robotic bin-picking. Thus, the term 'platform' is not rigorous.
Response: We appreciate the reviewer for their comment. While dataset generation is an important aspect of our research, it is only one part of the overall contribution. Our study also includes the development of VR modules and advancements in motion control algorithms, which collectively enhance the capabilities of the proposed system. Therefore, we have decided not to alter the title as it accurately reflects the multi-faceted nature of our research.
However, we have taken the reviewer's comment into consideration and made improvements to the relevant sections of the paper to better emphasize the contributions of the VR modules and motion control advancements. These enhancements aim to provide a comprehensive understanding of the research and highlight the significance of all the integrated components in advancing the field of robotic picking systems.
Once again, we appreciate the valuable feedback provided by the reviewer, and we believe that the updated sections effectively convey the breadth and depth of our contributions.
Comment/Suggestion #2: Some references for semantic or instance segementation should be cited, especially, some simulation-based dataset generation should be emphasized.
Response: We appreciate the reviewer's suggestion and have taken it into consideration. In response to the comment, we have included the following references in our manuscript to address the need for semantic or instance segmentation as well as simulation-based dataset generation:
- -T. Lee, C.-H. Tsai, and J.-Y. Chang, "A CAD-Free Random Bin Picking System for Fast Changeover on Multiple Objects." Proceedings of the ASME 2020 29th Conference on Information Storage and Processing Systems. ASME 2020 29th Conference on Information Storage and Processing Systems. Virtual, Online. June 24–25, 2020. V001T02A002. ASME.
- Qiao, Y. Zhu, H. Adam, A. Yuille, and L. -C. Chen, "ViP-DeepLab: Learning Visual Perception with Depth-aware Video Panoptic Segmentation," 2021 IEEE/CVF Conference on Computer Vision and Pattern Recognition (CVPR), 2021, pp. 3996-4007. [CrossRef]
- Shen, J. Yang, C. Wei, B. Deng, J. Huang, X. Hua, X. Cheng, and K. Liang, "DCT-Mask: Discrete Cosine Transform Mask Representation for Instance Segmentation," 2021 IEEE/CVF Conference on Computer Vision and Pattern Recognition (CVPR), 2021, pp. 8716-8725. [CrossRef]
- A. Ganea, B. Boom, and R. Poppe, "Incremental Few-Shot Instance Segmentation," 2021 IEEE/CVF Conference on Computer Vision and Pattern Recognition (CVPR), 2021, pp. 1185-1194. [CrossRef]
- Chen, Y. Li, T. Kong, L. Qi, R. Chu, L. Li, and J. Jia, "Scale-aware Automatic Augmentation for Object Detection," 2021 IEEE/CVF Conference on Computer Vision and Pattern Recognition (CVPR), 2021, pp. 9558-9567. [CrossRef]
- d. Geus, P. Meletis, C. Lu, X. Wen, and G. Dubbelman, "Part-aware Panoptic Segmentation," 2021 IEEE/CVF Conference on Computer Vision and Pattern Recognition (CVPR), 2021, pp. 5481-5490. [CrossRef]
- Yuan and R. C. Veltkamp, "PreSim: A 3D Photo-Realistic Environment Simulator for Visual AI," in IEEE Robotics and Automation Letters, vol. 6, no. 2, pp. 2501-2508, April 2021. [CrossRef]
Comment/Suggestion #3: Theoretical contributions are not outstanding. Only an introduction of a system is presented. The author should summarize some main contributions.
Response: We appreciate the valuable feedback provided by the reviewer. We have carefully considered their comment and made significant improvements to the abstract, introduction, and other sections of the paper to better highlight the unique contributions of our work.
While it is true that there are existing works in the field, we believe that our research brings several distinct advantages and advancements. Our paper introduces the RoboTwin metaverse platform, which combines a realistic virtual environment, physics engines, automated AI tools, robotic models, and virtual reality modules. This integrated approach sets our work apart from previous studies and offers several key benefits for vision-guided robotic picking systems.
Furthermore, our research addresses the challenge of rapid changeover in vision-guided robotic systems by utilizing a highly realistic metaverse scene. This approach enables rapid dataset generation, reducing the manual annotation process by 80 times. Additionally, our proposed system reduces simulation-to-real gap errors and provides a faster line changeover, meeting the requirements of flexible manufacturing systems.
We have expanded on these advantages and highlighted the unique aspects of our work in the updated sections of the paper. We believe that these improvements provide a clearer understanding of the significance and novel contributions of our research compared to existing works in the field.
Comments on the Quality of English Language: The form. English should be improved throughout the manuscript. The form is often too colloquial. The style is often too verbose. The authors are invited to carefully check the manuscript in this sense.
Response: We appreciate the reviewer's feedback and have conducted a thorough review of the manuscript to address the concerns regarding English form, colloquialism, and verbosity.
Reviewer 4 Report
This paper presents a solution for solving the challenges of using AI-based vision systems for robotic picking tasks in real-world factories, and then the proposed methods and system are verified by experiments. The full text is rich in content and has a relatively complete structure.
There are minor language errors that need to be checked and modified.
Author Response
Comment/Suggestion: This paper presents a solution for solving the challenges of using AI-based vision systems for robotic picking tasks in real-world factories, and then the proposed methods and system are verified by experiments. The full text is rich in content and has a relatively complete structure.
Reponse: Thank you very much for the recgonizition and encouragement. Your review efforts and comments are very much appreciated.
Comments on the Quality of English Language: There are minor language errors that need to be checked and modified.
Response: We thank the reviewer for their input. We have made adjustments to the revision accordingly.
Round 2
Reviewer 1 Report
This paper proposed the RoboTwin metaverse platform for vision-guided robotic random-bin-picking, which can apply the simulation results of the virtual environment to the real world. However, there are many similar works, and this paper has no obvious advantages compared with other works. Every sub-section of the RoboTwin system is not clearly explained, which makes the core content of this article not prominent enough. It is recommended that the author introduce the core innovation points of this article in detail, rather than briefly introduce all parts of the system. In addition, in order to further demonstrate the effectiveness of the proposed system, it is suggested that the authors provide some experimental videos.
Author Response
Comment/Suggestion point A: ...., there are many similar works, and this paper has no obvious advantages compared with other works.
Response: We appreciate the valuable feedback provided by the reviewer. We have carefully considered their comment and made significant improvements to the abstract, introduction, and other sections of the paper to better highlight the unique contributions of our work.
While it is true that there are existing works in the field, we believe that our research brings several distinct advantages and advancements. Our paper introduces the RoboTwin metaverse platform, which combines a realistic virtual environment, physics engines, automated AI tools, robotic models, and virtual reality modules. This integrated approach sets our work apart from previous studies and offers several key benefits for vision-guided robotic picking systems.
Furthermore, our research addresses the challenge of rapid changeover in vision-guided robotic systems by utilizing a highly realistic metaverse scene. This approach enables rapid dataset generation, reducing the manual annotation process by 80 times. Additionally, our proposed system reduces simulation-to-real gap errors and provides a faster line changeover, meeting the requirements of flexible manufacturing systems.
We have expanded on these advantages and highlighted the unique aspects of our work in the updated sections of the paper. We believe that these improvements provide a clearer understanding of the significance and novel contributions of our research compared to existing works in the field.
Furthermore, it is important to note that we have already applied for patent protection for the related automated image generation and annotation techniques in 2020. In late 2021, the international AI giant NVIDIA also released a similar feature called Replicator, although it is applied in a different domain. However, this serves as evidence that we are pioneers in this technology with technical depth and patent protection. This paper, on the other hand, focuses on the integration and application of this technology in the field of robotics, aiming to validate its effectiveness. Once again, we appreciate the reviewer's feedback and suggestions, which have helped us enhance the quality and clarity of our paper.
Comment/Suggesiton point B: Every sub-section of the RoboTwin system is not clearly explained, which makes the core content of this article not prominent enough. It is recommended that the author introduce the core innovation points of this article in detail, rather than briefly introduce all parts of the system.
Response: We would like to express our gratitude to the reviewer for their valuable comment. We apologize for any confusion caused by the lack of clarity in explaining the sub-sections of the RoboTwin system.
Based on the reviewer's suggestion, we have made significant revisions to the article to address this concern. We have provided more detailed explanations and highlighted the core innovation points of our research throughout the paper, emphasizing the unique contributions of each part of the RoboTwin system.
Specifically, we have expanded the introduction section to provide a comprehensive overview of the RoboTwin system and its key features, including the realistic virtual environment, physics engines, automated AI tools, robotic models, and virtual reality modules. We have also dedicated sections to describe the methodology, materials, and procedures used in the system, as well as presenting calibrated experimental results and engaging in discussions.
By providing more in-depth explanations and focusing on the core innovation points, we aim to make the core content of our article more prominent and ensure that readers have a clear understanding of the key contributions of our research.
We sincerely appreciate the reviewer's insightful suggestion, which has helped us improve the presentation and emphasis of our core innovation points.
Comment/Suggestion point #C: In addition, in order to further demonstrate the effectiveness of the proposed system, it is suggested that the authors provide some experimental videos.
Response: We thank the reviewer for their suggestion. We have added the following videos in the revision:
Video #1: https://www.youtube.com/watch?v=BuUc_1zTrrg
Video #2: https://www.youtube.com/watch?v=AnX3v12v8y4